# Simulation of the Biofiltration of Sulfur Compounds: Effect of the Partition Coefficients

Javier Silva [1,*] , Rodrigo Ortiz-Soto [1], Marcelo León [1] , Marjorie Morales [2] and Germán Aroca [3]

1   Escuela de Ingeniería Química, Pontifica Universidad Católica de Valparaíso, Av. Brasil 2162, Valparaíso 2340025, Chile; rodrigo.ortiz@pucv.cl (R.O.-S.); marcelo.leon@pucv.cl (M.L.)
2   Department of Energy and Process Engineering, Faculty of Engineering, Norwegian University of Science and Technology (NTNU), NO-7491 Trondheim, Norway; marjorie.morales@ntnu.no
3   Escuela de Ingeniería Bioquímica, Pontificia Universidad Católica de Valparaíso, Av. Brasil 2085, Valparaíso 2374631, Chile; german.aroca@pucv.cl
*   Correspondence: javier.silva@pucv.cl; Tel.: +56-32-2372618

**Abstract:** The effect of the partition coefficient on the simulation of the operation of a biotrickling filter treating a mixture of sulfur compounds was analyzed to evaluate the pertinence of using Henry's law in determining its removal capacity. The analysis consisted of the simulation of a biotrickling filter that bio-oxides hydrogen sulfide ($H_2S$), dimethyl sulfide (DMS), methyl mercaptan (MM) and dimethyl disulfide (DMDS) using different types of models for determining the partition coefficient: Henry's law for pure water, Henry's law adjusted from experimental data, a mixed model (Extended UNIQUAC) and a semi-empirical model of two-parameters. The simulations were compared with experimental data. It was observed that Henry's law for pure water could produce significant deviations from empirical data due to the liquid phase not being pure water. The two-parameter model better fits with similar results compared to the extended UNIQUAC model, with a lower calculation cost and necessary parameter amount. It shows that semi-empirical models can considerably improve simulation accuracy where complex phase interactions are present.

**Keywords:** partition coefficient; culture medium; biotrickling filter; activity coefficient; modeling





## 1. Introduction

Total reduced sulfurs (TRS) are compounds emitted by several processes such as kraft pulp mills, wastewater treatment plants, and oil refineries. They are well-known for their odor impact and adverse health effects [1,2]. The population can perceive such compounds at minimum concentrations, causing a nuisance to people living near production plants where these compounds are present [3,4]. $H_2S$, dimethyl sulfide (DMS), dimethyl disulfide (DMDS), and methyl mercaptan (MM) are examples of these compounds. Several authors have shown that biotrickling filters efficiently remove those compounds from such emissions [5].

Several models have been developed to predict the removal capacities of biotrickling filters at different modalities to determine optimal operating conditions [6–8]. These models require several parameters related to transport, kinetics, support characteristics and biofilm. Silva et al. [9] showed that the partition coefficient is one of the most influential parameters in simulating a biotrickling filter to remove $H_2S$.

From the operational point of view, a better removal capacity is obtained when the partition coefficient decreases for the case of volatile organic compounds (VOC), except when there is a limitation in oxygen availability, evidencing this parameter's importance in the simulation of biotrickling filters [10].

Many models use Henry's constant as a partition coefficient to describe the distribution of gas–liquid species in biofilters [11–13]. Its use assumes a constant relationship between the concentrations in both phases; the liquid phase is pure water, and the gaseous phase is

very diluted [14]. Hence, its use could involve high deviations compared to real equilibrium data [15]. Mixed models, such as extended UNIQUAC, emerge as an improved way of describing these systems [16]. Despite the promising results of these models, the high amount of parameters required and the high cost of calculation means their use is not extended [17–19].

Semi-empirical models have emerged as a reliable alternative to modeling and simulating complex liquid–vapor systems, especially when there are many species in the liquid phase [20,21]. Many semi-empirical models are present in the literature [21]. These models have the advantage of describing complex systems where the species' interactions are not completely clear [22,23]. Lee et al. [24] developed a simplified semi-empirical model of two parameters to describe ionic solutions, obtaining good results for many substances in vapor–liquid equilibria.

In this work, we report the effect of the gas–liquid partition coefficient obtained from different models in the simulation of a biotrickling filter treating air containing low concentrations of $H_2S$, MM, DMS and DMDS. The simulations are compared with experimental data.

## 2. Materials and Methods

### 2.1. Biotrickling Filter

A biotrickling filter was set up using a polyvinyl chloride (PVC) column 40 cm high and 6.5 cm in diameter, filled with polypropylene rings 1 cm in diameter, with a density of 1.02 kg $L^{-1}$ in a volume of 1.3 L, and with a specific surface of 2.3 $m^2$ $L^{-1}$. This column had five equidistant sampling points along its height.

*Thiobacillus thioparus* ATCC 23645 was used to inoculate the support. This microorganism was propagated in liquid medium No. 290 (ATCC) using sodium thiosulphate (10 g $L^{-1}$) as an energy source and was incubated at 30 °C and 200 rpm in a mechanical shaker. The medium composition (in g $L^{-1}$) was $Na_2HPO_4 \cdot 7H_2O$ 2.27; $KH_2PO_4$ 1.8; $MgCl_2 \cdot 7H_2O$ 0.1; $(NH_4)_2SO_4$ 1.98; $MnCl_2 \cdot H_2O$ 0.023; $CaCl_2$ 0.03; $FeCl_3 \cdot 6H_2O$ 0.033; $Na_2CO_3$ 1; $Na_2S_2O_3 \cdot 5H_2O$ 15.69 and pH was adjusted to 6.8.

The rings were inoculated with 1 L of an active culture containing a cell concentration of $1.5 \cdot 10^{10}$ cell $L^{-1}$. The cell suspension was recirculated for two days through the packed column to adsorb the microorganisms. After the biofilm was formed, ATCC 290 culture medium without thiosulfate was recirculated through the column using a spray at the top. Every two days, 0.5 L of the solution was replaced by a new solution to maintain the pH at 6.8 and the sulfate concentration below 10 g $L^{-1}$. The number and viability of the cells in the biofilm were monitored using an epifluorescence microscope (Eclipse model, Nikon). The biomass attached to the polyethylene rings was released using ultrasound (43 kHz for 5 min) and was suspended in 10 $cm^3$ of sterile medium ATCC 290 without thiosulfate.

Figure 1 shows a diagram of the Experimental System.

$H_2S$ was generated by the reaction between a sodium sulfide solution ($Na_2S$, 1–3%) and a hydrochloric acid solution (HCl, 0.5 N) using a specially designed apparatus [25]. MM was generated by diluting known volumes of an incoming flow with 99% MM with air. The DMS was produced using a capillary diffusion system described in [26]. DMDS was generated by a convection system where an air stream vertically impacts the DMDS surface in a 1 cm diameter tube equipped with a gas outlet near the system's top. The temperature was controlled at 30 °C using a thermostatic bath.

The adaptation was carried out by feeding each substance separately; 1 ppm $\pm$ 0.04 ppm of each substance separately until the steady-state was reached. Before each adaptation, the system was fed using 4, 10, and 20 ppm of each compound to avoid substantially modifying the liquid phase's ionic strength. The culture media was recirculated in the column with a flow of 0.6 $cm^3$ $s^{-1}$ while the air flow rate was 0.5 L $min^{-1}$. The empty residence time was 120 s. The flow of the gas stream was regulated using direct-reading rotameters fitted with precision valves. The nutrient solution was fed in countercurrent to

the gas and was received in a closed vessel. This solution was then recirculated back to the biofiltration column.

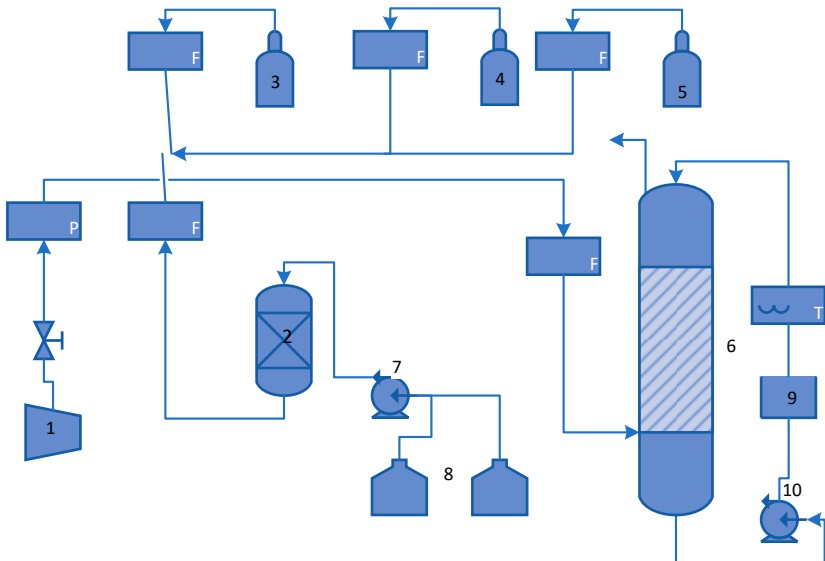

**Figure 1.** Experimental system for oxidation of sulfur compounds in the biotrickling filter. (1) Compressor; (2) $H_2S$ generator; (3) DMDS generator; (4) MM cylinder; (5) DMS generator; (6) Biotrickling filter; (7) pump $H_2S$ generator; (8) tanks $Na_2S$ and HCl for $H_2S$ production; (9) recipient of the nutrient solution with thermostatic bath; (10) pump of recirculating media.

The gas-phase concentrations were determined using a gas chromatograph (Perkin Elmer Clarus 500, Shelton, CT, USA). This equipment included a packed column Supelpack S (Supelco) with a photometric flame detector (FPD) using helium as the carrier gas with a minimum detectable quantity of $3 \times 10^{-12}$ g C s$^{-1}$ octane.

### 2.2. Obtaining Vapor-Liquid Equilibrium Data

An equilibrium system was implemented through a liquid–vapor batch system to determine the distribution of the species and fit the empirical parameters by introducing known quantities of each gas. The concentration in the gas phase was measured when these substances were fed, and the equilibrium was reached. The concentration in the liquid phase was determined through a mass balance (Equation (1)).

$$c_L = \frac{P \cdot PM \cdot V_g}{R \cdot T \cdot V_L \cdot d} \left( c_F - c_g \right), \tag{1}$$

where $c_L$ is the concentration in the liquid phase (ppm), P is the system pressure (atm), PM is the molecular weight (g mol$^{-1}$), R is the universal constant of the ideal gases (0.082 atm L mol$^{-1}$ K$^{-1}$), T is the system temperature (K), $V_L$ is the volume of liquid (L), d is the density of the liquid phase (g L$^{-1}$), $c_F$ is the concentration of the feed compound (ppm), $c_g$ is the concentration of the mixture in the gaseous phase in equilibrium (ppm), and $V_g$ is the volume of the gaseous phase of the system (L). A complete list of every symbol used in this work is displayed in Appendix A.

A known amount of liquid medium of thiosulfate ATCC 290 (50 ± 0.1 cm$^3$) was introduced into a balloon of measured volume (130 ± 0.1 cm$^3$), and was pH adjusted to 6.8. The pressure inside was determined using a manometer connected to it. A vial was sealed hermetically, and a known amount of $H_2S$, DMS, MM, or DMDS was added in separate experiments. The vial was immersed in a thermostatic bath at 50 ± 0.1 °C. Once the equilibrium was reached, which was appreciated when the pressure became constant (±0.02 atm$_g$), samples of 1 ± 0.1 cm$^3$ were taken at different times to verify that the equilibrium was achieved.

### 2.3. Mathematical Model

2.3.1. Biotrickling Filter Model

The biotrickling filter was simulated using a model [27] that considers mass transfer and biochemical oxidation (Equation (2)). This model considers the biotrickling filter as a packed column where the microorganism forms a biofilm on the support material. When air, which contains a particular substance, flows through the column, it transfers this substance from the gas phase to the liquid phase, diffusing to the biofilm being oxidized by microbial activity.

The model considers the following equations:

General mass balance to the gas phase:

$$\frac{1}{Pe}\cdot\frac{\partial^2 C_g}{\partial\zeta^2} - \frac{\partial C_g}{\partial\zeta} + v_b\cdot Ti\cdot\frac{\partial C_b}{\partial\psi}\Big|_{\psi=0} = 0, \tag{2}$$

where $C_g$ is the dimensionless concentration in the gas phase, $C_b$ is the dimensionless concentration in the biofilm, $\zeta$ is the dimensionless axial coordinate along the bed, Pe is the Péclet mass transfer number, $v_b$ is the specific volume of the biofilm, Ti is the ratio between residence and diffusion time, and $\psi$ is the dimensionless spatial ordinate in the biofilm.

The general mass balance of the biofilm is described in Equation (3).

$$\frac{\partial^2 C_g}{\partial\psi^2} - Th^2\cdot\phi_b\cdot\kappa = 0, \tag{3}$$

where Th is the Thiele module, $\kappa$ is the dimensionless speed of specific degradation velocity, and $\phi_b$ is the dimensionless biomass concentration.

The kinetic equation for the velocity of dimensionless degradation is expressed in Equation (4).

$$\kappa = \frac{1}{Y_{X/S}}\frac{C_b}{\sigma + C_b}. \tag{4}$$

$Y_{X/S}$ is the yield coefficient of biomass over the TRS (g biomass g$^{-1}$ TRS), and $\sigma$ is the dimensionless Monod constant.

The dimensionless concentration of the pollutant in the gas phase, the dimensionless concentration in the biofilm, and the dimensionless concentration of biomass in the biofilm are defined in Equations (5)–(7), respectively.

$$C_g = \frac{c_g}{c_g^{in}} \tag{5}$$

$$C_b = \frac{c_b}{c_b^{in}} \tag{6}$$

$$\phi_b = \frac{X_b}{c_g^{in}}, \tag{7}$$

where $c_g$ is the contaminant concentration in the gas phase (g m$^{-3}$), $c_g^{in}$ is the contaminant input concentration in the gas phase (g m$^{-3}$), $c_b$ is the contaminant concentration in the biofilm, $c_b^{in}$ is the contaminant concentration in the biofilm at the biofilm input (g m$^{-3}$) and $X_b$ is the biomass concentration in the biofilm.

The dimensionless spatial coordinate and axial ordinate in the biofilm are defined in Equations (8) and (9), respectively:

$$\psi = \frac{r}{\delta} \tag{8}$$

$$\zeta = \frac{z}{H}, \tag{9}$$

where $\delta$ is the thickness of the biofilm (m), $r$ is the spatial ordinate in the biofilm (m), $z$ is the axial coordinate (m), and H is the height of the biofilter (m).

The dimensionless parameters Pe (modified Péclet number), $v_b$ (specific biofilm volume), Ti (diffusion ratio residence time), and Th (modified thiele module) are described in particular in Equations (10)–(13).

$$Pe = \frac{V_z \cdot H}{W} \tag{10}$$

$$v_b = \delta \cdot \alpha_i \cdot a_s \tag{11}$$

$$Ti = \frac{\frac{H}{V_z}}{\frac{\delta^2}{D_b}} = \frac{\tau_R}{\tau_D} \tag{12}$$

$$Th = \sqrt{\frac{\delta^2 \cdot \mu_{max}}{D_b}}, \tag{13}$$

where $a_s$ is the specific surface per unit volume of the reactor $(m^{-1})$, $V_z$ is the velocity of the gas $(m\ s^{-1})$, W is the coefficient of dispersion $(m^2\ s^{-1})$, $\alpha$ is the fraction of the support surface covered by biofilm, $\mu_{max}$ is the specific velocity of growth $(s^{-1})$, $\tau_R$ is the residence time of the gas (s), and $\tau_D$ is the characteristic time of diffusion in the biofilm (s).

The nondimensional Monod constant is defined in Equation (14).

$$\sigma = \frac{K_s}{c_b^{in}}, \tag{14}$$

where $K_s$ is the constant of Monod (ppm).

The boundary conditions for the model for gas inlet, gas outlet, external biofilm surface, and biofilm contact with support rings are reported in Equations (15)–(18), respectively.

$$Z = 0;\ 0 \le \psi \le 1: -\frac{\partial C_g}{\partial \zeta} + v_b \cdot Ti \cdot \left.\frac{\partial C_b}{\partial \psi}\right|_{\psi=0} = 0 \tag{15}$$

$$\zeta = 1;\ 0 \le \psi \le 1: -\frac{\partial C_g}{\partial \zeta} + v_b \cdot Ti \cdot \left.\frac{\partial C_b}{\partial \psi}\right|_{\psi=0} = 0 \tag{16}$$

$$\psi = 0;\ 0 \le \zeta \le 1: c_b = \frac{c_g}{m} \tag{17}$$

$$\psi = 1;\ 0 \le \zeta \le 1: \frac{\partial C_{bi}}{\partial \psi} = 0, \tag{18}$$

where m is the partition coefficient.

2.3.2. Equations for the Determining the Partition Coefficient

One of the simplest models representing the liquid–vapor partition coefficient corresponds to Henry's law (Equation (19)).

$$M_i = H_i = \frac{c_b}{y_i P}, \tag{19}$$

where $m_i$ is the partition coefficient, $H_i$ is Henry's constant $(M\ atm^{-1})$, $y_i$ is the gas phase composition, P is the pressure, and $c_b$ is the liquid phase composition (M).

The activity coefficient can modify this model to improve its accuracy (Equation (20)).

$$M_i = H_i = \frac{c_b \cdot \gamma_i}{y_i P}, \tag{20}$$

where $\gamma_i$ is the activity coefficient.

The UNIQUAC model is useful for multicomponent non-electrolytic solutions to determine the activity coefficient [28]. For mixed solutions, it is possible to use the extended UNIQUAC model [29] as the combination of the UNIQUAC model and the Debye–Hückel model (Equation (21)).

$$\ln(\gamma_i) = \ln(\gamma_i)^U + \ln(\gamma_i)^{DH},\tag{21}$$

where $\ln(\gamma_i)^U$ indicates the contribution of the UNIQUAC model to the activity coefficient of the compound, and $\ln(\gamma_i)^{DH}$ indicates the contribution of the Debye–Hückel model of the same compound in the mixture. The activity coefficient is described by the UNIQUAC model as indicated by Equations (22)–(26).

$$\ln(\gamma_i)^U = \ln\frac{\Phi_i}{x_i} + 5q_i\ln\frac{\theta_i}{\Phi_i^*} + l_i - \frac{\Phi_i}{x_i}\sum_{j=1}^{n} x_j l_j - q_i\ln\left(\sum_{j=1}^{n}\theta_j\tau_{ij}\right) + q_i - q_i\sum_{j=1}^{n}\frac{\theta_j\tau_{ij}}{\sum_{k=1}^{n}\theta_k\tau_{kj}}\tag{22}$$

$$\Phi_i = \frac{x_i r_i}{\sum_j x_j r_j}\tag{23}$$

$$\theta_i = \frac{x_i q_i}{\sum_j x_j q_j}\tag{24}$$

$$\tau_{ij} = e^{\left(-\frac{u_{ji}-u_{ii}}{RT}\right)}\tag{25}$$

$$l_i = 5(r_i - q_i) - (r_i - 1),\tag{26}$$

where $q_i$ is the relative surface area, $r_i$ is the relative molecular volume, and $u_{ij}$ is an adjustable interaction parameter.

The Debye–Hückel equation (Equation (27)) can be represented by simplifying the higher-order terms of the complete expression according to the following assumptions:

- Complete dissociation of electrolytes;
- Oppositely charged ions surround each ion;
- Low electrolyte concentrations.

$$\ln(\gamma_i)^{DH} = \frac{-Z_i^2 A\sqrt{I}}{1 + d_i B\sqrt{I}}.\tag{27}$$

Equation (28) shows the ionic strength of the solution.

$$I = \frac{1}{2}\sum_i c_i Z_i^2,\tag{28}$$

where $I$ is the ionic strength of the solution (M), $Z$ is the charge of each element, $c_i$ is the molar concentration of each compound in the solution (M), $d_i$ is the ionic radius, while $A$ and $B$ are adjustable model parameters.

A truncated model for the $A$ parameter is considered [30] in Equation (29), where the value of $d_i B$ from the Debye–Hückel model can be estimated as 1.5 $(\text{kg mol})^{1/2}$, and the value of $A$.

$$A = 1.131 + 1.335 \cdot 10^{-3}(T - 273.15) + 1.164 \cdot 10^{-5}(T - 273.15)^2.\tag{29}$$

The complexity of such models has promoted the development of simplified semi-empirical models. Lee et al. [24] developed a two-parameters model (Equation (30)) that depends on a proportional factor ($\alpha$) and the effective radius of an ionic sphere ($\beta$), which are determined through fitting experimental data.

$$\ln(\gamma) = -\frac{\beta}{\alpha}\cdot I^{-\frac{1}{2}}\left(\frac{1 - \kappa_i\alpha}{1 + \kappa_i\alpha}\right),\tag{30}$$

where $\gamma$ is the activity coefficient, I is the ionic force (Equation (28)), and $\kappa$ is the Debye screen length ($\kappa_i{}^{-1}$ = 3.0434 Å·$I^{-1/2}$·m). That model is capable of showing good fits in dilute ionic solutions.

In this work, the effect of different partition coefficient models is analyzed in the simulation of a biotrickling filter, considering Henry's law for pure water, Henry's law by adjusting experimental data, the theoretical extended UNIQUAC model, and a semi-empirical model of two-parameters [24].

### 2.3.3. Simulation

The model was solved by a second central order finite difference and a MatLab algorithm [31]. The parameters applied for model resolution are shown in Table 1.

**Table 1.** Parameters used in simulations.

| Parameter | Value | Units | Reference |
|---|---|---|---|
| Henry's constant $H_2S$ | $1 \times 10^{-1}$ | M atm$^{-1}$ | [32,33] |
| Henry's constant DMS | $4.8 \times 10^{-1}$ | M atm$^{-1}$ | [32,33] |
| Henry's constant MM | $7.1 \times 10^{-1}$ | M atm$^{-1}$ | [32,33] |
| Henry's constant DMDS | $9.3 \times 10^{-1}$ | M atm$^{-1}$ | [32,33] |
| Diffusion coefficient of $H_2S$ in water | $1.93 \times 10^{-9}$ | m$^2$ s$^{-1}$ | [34] |
| Diffusion coefficient of DMS in water | $1.51 \times 10^{-9}$ | m$^2$ s$^{-1}$ | [35] |
| Diffusion coefficient of MM in water | $1.71 \times 10^{-9}$ | m$^2$ s$^{-1}$ | [36] |
| Diffusion coefficient of DMDS in water | $1.08 \times 10^{-9}$ | m$^2$ s$^{-1}$ | [37] |
| Maximum specific growth velocity $H_2S$ | 0.045 | h$^{-1}$ | [38] |
| Maximum specific growth velocity DMS | 0.004 | h$^{-1}$ | [39] |
| Maximum specific growth velocity MM | 0.009 | h$^{-1}$ | [39] |
| Maximum specific growth velocity DMDS | 0.008 | h$^{-1}$ | [40] |
| Saturation constant $H_2S$ | 84.7 | ppm | [37] |
| Saturation constant DMS | 4.8 | ppm | [38] |
| Saturation constant MM | 17.7 | ppm | [39] |
| Saturation constant DMDS | 7.1 | ppm | [40] |
| Yield $H_2S$ | 0.03 | Nondimensional | [37] |
| Yield DMS | 0.05 | Nondimensional | [38] |
| Yield MM | 0.33 | Nondimensional | [39] |
| Yield DMDS | 0.98 | Nondimensional | [40] |

For the extended UNIQUAC model, the liquid phase was considered an ionic solution composed of different dissolved and ionized substances. The reactions of dissolution considered are shown in Appendix B.

The parameters needed for the UNIQUAC model were estimated according to the method developed by Pahlevanzade and Mohseni-Ahooei [41], using as a reference the parameters utilized by Arrad et al. [42–44], Boulkroune et al. [45], Raatikainen and Laaksonen [46] and Thompsen et al. [47].

The values of Henry's constant for the biotrickling liquid phase and parameters $\alpha$ and $\beta$ were obtained by adjustment according to data collected by the experimental system indicated above.

Finally, simulations were carried out to determine each model's effect in evaluating the biofilter's removal capacity.

### 2.4. Statistical Methods

A one-factor analysis of variance (ANOVA) was assessed with a 10% significance level of each model's residuals [48] to compare the models studied. Due to the different initial conditions, a blocking effect of the sampling height was considered. The ANOVA results were obtained by Equation (31).

$$p\text{-Value} = P(F > f_{0,a,b}) = \int_{f_{0,a,b}}^{\infty} f(x)dx, \tag{31}$$

where f(x) is Fisher's probability distribution with a degrees of freedom in the numerator (models used in simulations) and b degrees of freedom in the denominator (experiment error after blocking), $f_{0,a,b}$ is the variance ratio between models residuals and statistical experiment error. If the *p*-Value is lower than the significance level, a difference between residuals among results of any pair of models is demonstrated. The differences between models were identified by calculating the confidence intervals of the mean residual difference for each possible couple of models in each experiment, as is shown in Equation (32).

$$\mu_i - \mu_j = RMSE_i - RMSE_j \pm t_{5\%,9} \cdot \sqrt{2 \cdot \frac{MSE_{exp}}{4}} = \Delta RMSE \pm LSD, \tag{32}$$

where $\mu_i - \mu_j$ is the real difference between the residuals. $RMSE_i$ and $RMSE_j$ are the roots of mean square errors of each model, respectively, in each case, and $t_{5\%,9}$ is the appropriate two-tailed t-Student statistic for this ANOVA. In cases where the magnitude of the difference of the root of the mean square error ($\Delta RMSE$) of each model is higher than its least significant difference (LSD), it is ensured that its residuals are different. In these cases, if $\Delta RMSE$ is positive, model i has higher residuals than model j.

## 3. Results

The values for Henry's constants adjusted and the parameters α and β obtained for each studied compound are shown in Table 2. These results have the same magnitude order as the values obtained by Lee et al. [24] for their compound studied.

**Table 2.** Parameters α and β for the two-parameter model.

| Parameter | H$_2$S | DMS | MM | DMDS |
|---|---|---|---|---|
| $H_i \cdot 10^{-1}$ | 3.1 | 6.9 | 8.0 | 12.9 |
| $\alpha \cdot 10^{-10}$ | 0.07 | 1.42 | 0.86 | 1.47 |
| $\beta \cdot 10^{-10}$ | 1.51 | 1.81 | 2.27 | 1.36 |

Table 3 shows the removal capacities obtained from each model studied and its comparison with the experimental results.

**Table 3.** Removal capacities determined by experimental data (Exp), Henry's Law (HL), Henry's law adjusted (HA), Extended UNIQUAC (EU), Two-parameters model (T).

| Compound | Inlet Concentration (ppm) | Exp | HL | HA | UE | TM |
|---|---|---|---|---|---|---|
| | 4 | 98% | 95% | 98% | 100% | 100% |
| H$_2$S | 10 | 98% | 95% | 95% | 99% | 99% |
| | 20 | 96% | 94% | 95% | 94% | 98% |
| | 4 | 70% | 64% | 65% | 63% | 67% |
| DMS | 10 | 28% | 28% | 29% | 27% | 28% |
| | 20 | 15% | 14% | 16% | 14% | 14% |
| | 4 | 88% | 90% | 91% | 80% | 91% |
| MM | 10 | 80% | 82% | 83% | 80% | 81% |
| | 20 | 70% | 63% | 65% | 74% | 72% |
| | 4 | 88% | 80% | 81% | 80% | 86% |
| DMDS | 10 | 35% | 44% | 47% | 36% | 36% |
| | 20 | 23% | 24% | 26% | 22% | 21% |

Regarding each model prediction capacity for experimental TRS removal, it can be seen that higher differences are obtained with Henry's Law, with a different average of 4% and a maximum difference of 9% in the case of DMDS at 10 ppm, overestimating its removal. The adjusted Henry's Law also presents an average of 4% difference, and in the same DMDS case, its removal was overestimated by 12%. For extended UNIQUAC simulations, its difference average is 3%, and a maximum difference is observed for the

same compound but at 4 ppm, underestimating the removal by 8%. For the two-parameter model, the difference average is the lowest analyzed (2%), and the maximum difference was observed in the case of the lower inlet concentrations of DMS and MM, with a 3% underestimation and 3% of overestimation, respectively.

Figures 2–4 compare the experimental results and those obtained in simulations at 4, 10, and 20 ppm of input for each gas studied.

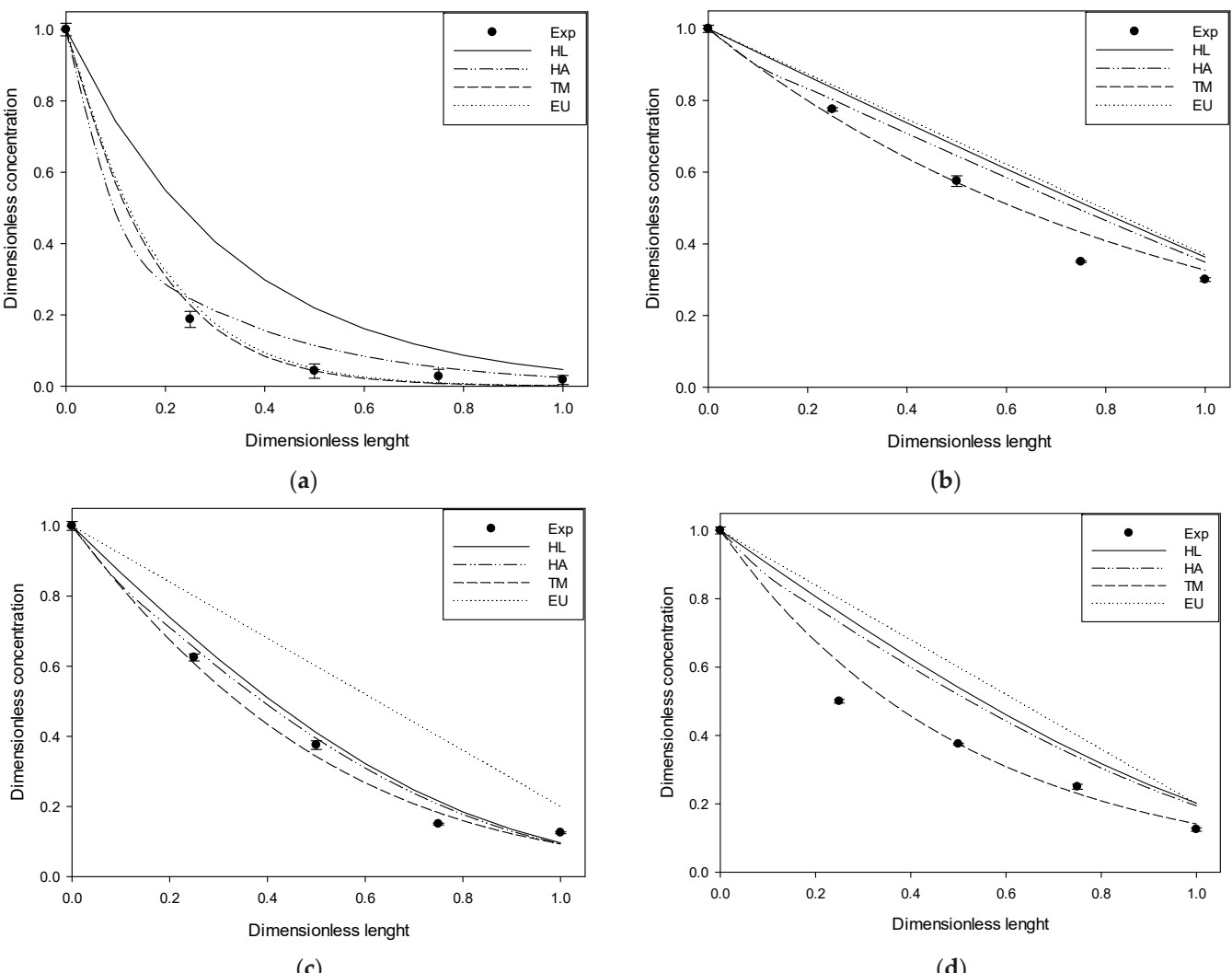

**Figure 2.** Nondimensional simulation results compared to experimental data with 4 ppm of input. (**a**) $H_2S$, (**b**) DMS, (**c**) MM, (**d**) DMDS.

The figures above show the behavior of the dimensionless profiles along the column. In the $H_2S$ case, an important removal was obtained in the first part of the column. The DMS had a linear behavior across the column length versus its removal. MM and DMDS have similar forms to $H_2S$ and DMS, respectively. On the other hand, it is possible to observe by visual inspection how the use of Henry's law for pure water deviates significantly from the experimental data. The adjusted Henry's Law, although closer, follows, in general, the same form as Henry's Law for pure water. Better behavior is observed in the case of the extended UNIQUAC model and the two-parameter model, generally having a closer fit to the experimental data. At high input concentration, it is possible to observe some deviation in the two-parameter model, possibly attributable to an increase in the solution's ionic strength, which was considered constant in the model.

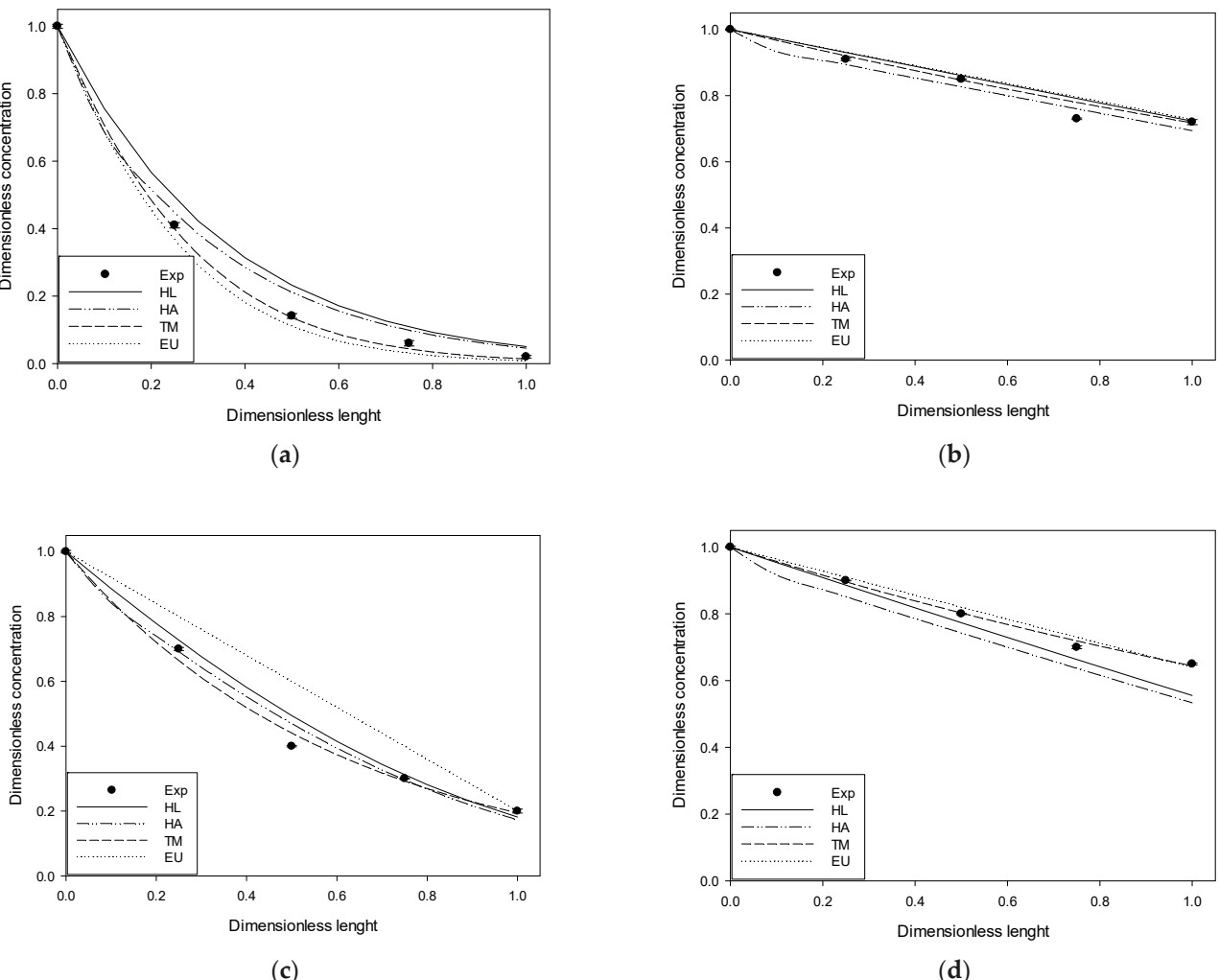

**Figure 3.** Nondimensional simulation results compared to experimental data with 10 ppm of input. (**a**) $H_2S$, (**b**) DMS, (**c**) MM, (**d**) DMDS.

Table 4 shows the values of the determination coefficients $R^2$ of each model for each input concentration and substance. The results indicate that, in the $H_2S$ case, the two-parameter model had better adjustment than the other models at all concentrations. Henry's law had a lower fit than other models in every input load analyzed. For DMS, the adjusted Henry's Law shows a better fit than the other simulations at 10 and 20 ppm, but for 4 ppm, the two-parameter model offers a better description of the process. On the other hand, MM gets its best description with the two-parameter model in every concentration, as for DMDS in 4 and 10 ppm.

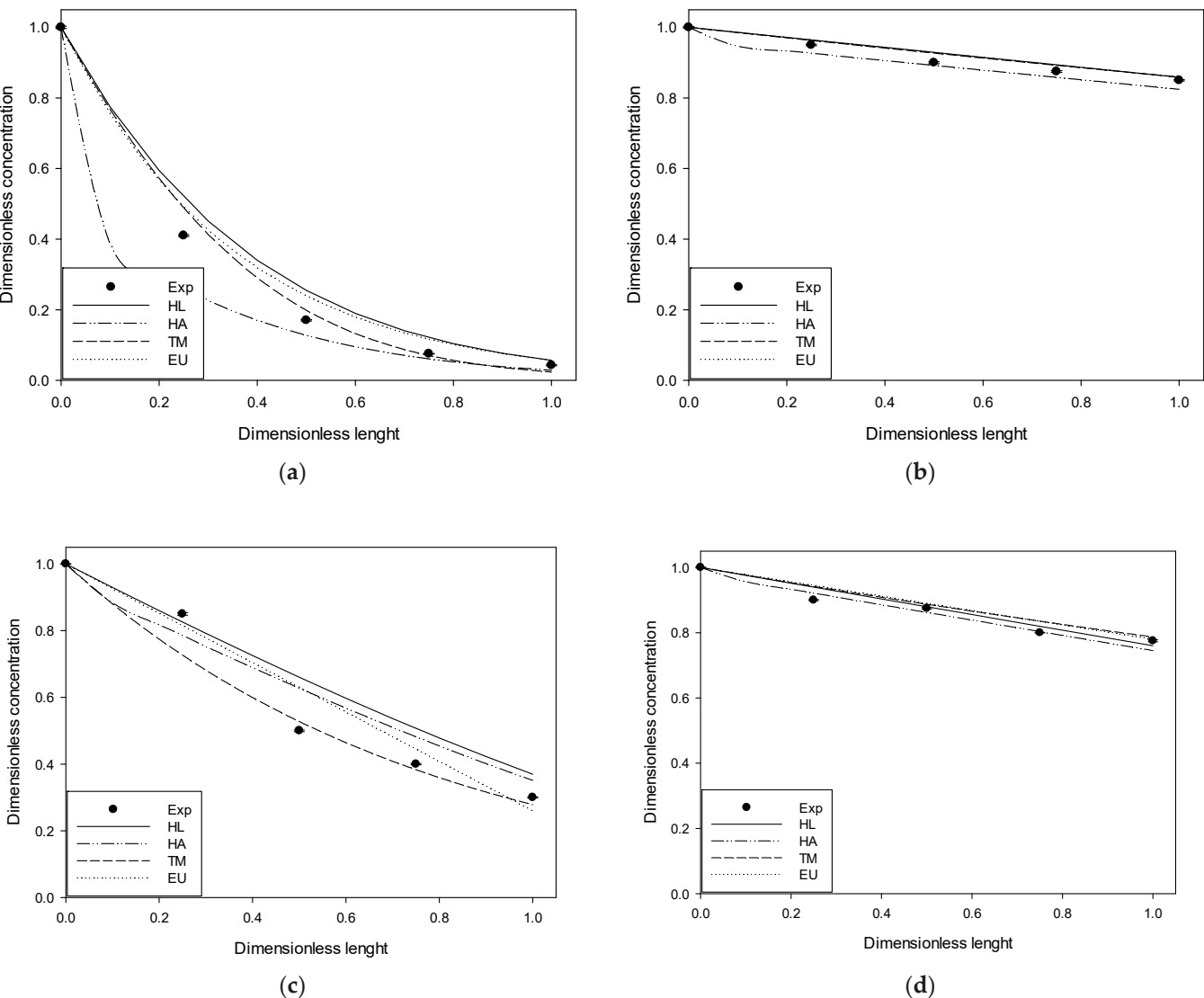

**Figure 4.** Nondimensional simulation results compared to experimental data with 20 ppm of input. (**a**) $H_2S$, (**b**) DMS, (**c**) MM, (**d**) DMDS.

**Table 4.** Determination coefficients $R^2$ (%).

| Species\Concentration | Model | 4 ppm | 10 ppm | 20 ppm |
|---|---|---|---|---|
| $H_2S$ | HL | 83.40 | 97.28 | 96.62 |
| | HA | 98.74 | 98.74 | 98.02 |
| | EU | 99.54 | 99.42 | 97.85 |
| | TM | 99.71 | 99.92 | 98.83 |
| DMS | HL | 87.12 | 92.35 | 89.67 |
| | HA | 91.61 | 95.99 | 98.55 |
| | EU | 84.56 | 91.07 | 89.95 |
| | TM | 97.75 | 95.48 | 91.31 |
| MM | HL | 98.30 | 97.64 | 88.23 |
| | HA | 99.02 | 98.67 | 91.77 |
| | EU | 72.04 | 85.98 | 93.99 |
| | TM | 99.36 | 99.29 | 95.34 |
| DMDS | HL | 75.64 | 86.39 | 93.05 |
| | HA | 81.20 | 71.53 | 95.21 |
| | EU | 63.01 | 98.17 | 88.98 |
| | TM | 97.08 | 99.49 | 89.71 |

Usually, adjusted Henry's Law was the best fit of data after the two-parameter model. The two-parameter model typically had the second-best performance when it was the best of all representations. On the other hand, it is observed that the Extended UNIQUAC model does not have the best fit in any case, and half of the studied models have worse performance than Henry's Law.

## 4. Discussion

Regarding the removal capacity of the system in Table 2, it is possible to observe that a high removal of $H_2S$ was obtained under the same operating conditions. These results demonstrate the preference of this microorganism for simpler metabolites as an energy source [49], which involve lower energy consumption because the metabolization of more complex substances requires higher energy consumption in the form of NADH [50].

Observing determination coefficients in Table 4, the difference in $R^2$ for each model is not visibly different in each experiment. Values of $R^2$ higher than 60% are considered acceptable and, as a first analysis, all models would be adequate [51]; nonetheless, visual trends suggest a difference in the representation of the phenomenon at low input concentrations.

A variance analysis (ANOVA) of the residuals of each model was performed to confirm any differences between models. Due to the models having different initial conditions, a blocking effect of the sampling height was considered. Table 5 shows the *p*-values for each case to determine the difference in the behavior of the models.

**Table 5.** *p*-values between models at different input concentrations per compound.

| Inlet Concentration (ppm) | $H_2S$ | DMS | MM | DMDS |
|:---:|:---:|:---:|:---:|:---:|
| 4 | 2.75% | 0.04% | 0.07% | 0.05% |
| 10 | 0.48% | 5.14% | 3.65% | 0.77% |
| 20 | 6.22% | 1.05% | 54.80% | 69.90% |

Supposing a 10% significance level, it is possible to observe a difference between the models studied at low and medium input concentrations but not at high levels for MM and DMDS.

Table 6 shows the results of calculating the confidence interval of the mean residual difference for each possible pair of models in each experiment.

**Table 6.** Mean Square Error Root Difference (ΔRMSE) and Least Significant Difference (LSD) for each model and compound at different concentrations.

| Species | Inlet Concentration | 4 ppm | | 10 ppm | | 20 ppm | |
|:---:|:---:|:---:|:---:|:---:|:---:|:---:|:---:|
| | Model i-j | ΔRMSE | LSD | ΔRMSE | LSD | ΔRMSE | LSD |
| $H_2S$ | HL-HA | 0.1011 | | 0.0200 | | 0.0142 | |
| | HL-EU | 0.1180 | | 0.0337 | | 0.0114 | |
| | HL-TM | 0.1227 | 0.0834 | 0.0522 | 0.0237 | 0.0305 | 0.0215 |
| | HA-EU | 0.0169 | | 0.0137 | | −0.0028 | |
| | HA-TM | 0.0217 | | 0.0322 | | 0.0163 | |
| | EU-TM | 0.0047 | | 0.0184 | | 0.0191 | |
| DMS | HL-HA | 0.0238 | | 0.0076 | | 0.0116 | |
| | HL-EU | −0.0099 | | −0.0035 | | 0.0003 | |
| | HL-TM | 0.0636 | 0.0249 | 0.0077 | 0.0092 | 0.0014 | 0.0068 |
| | HA-EU | −0.0337 | | −0.0111 | | −0.0113 | |
| | HA-TM | 0.0398 | | 0.0001 | | −0.0102 | |
| | EU-TM | 0.0735 | | 0.0112 | | 0.0011 | |

**Table 6.** *Cont.*

| Inlet Concentration | | 4 ppm | | 10 ppm | | 20 ppm | |
|---|---|---|---|---|---|---|---|
| Species | Model i-j | ΔRMSE | LSD | ΔRMSE | LSD | ΔRMSE | LSD |
| MM | HL-HA | 0.0121 | | 0.0101 | | 0.0089 | |
| | HL-EU | −0.1358 | | −0. 0628 | | 0.0278 | |
| | HL-TM | 0.0168 | 0.0601 | 0.0151 | 0.0548 | 0.0427 | 0.0707 |
| | HA-EU | −0.1478 | | −0.0724 | | 0.0189 | |
| | HA-TM | 0.0048 | | 0.0050 | | 0.0338 | |
| | EU-TM | 0.1526 | | 0.0774 | | 0.0149 | |
| DMDS | HL-HA | 0.0185 | | −0.0288 | | 0.0029 | |
| | HL-EU | −0.0366 | | 0.0254 | | −0.0053 | |
| | HL-TM | 0.1136 | 0.0495 | 0.0350 | 0.0331 | −0.0054 | 0.0188 |
| | HA-EU | −0.0552 | | 0.0542 | | −0.0082 | |
| | HA-TM | 0.095 | | 0.0638 | | −0.0083 | |
| | EU-TM | 0.1503 | | 0.0096 | | −0.0001 | |

It can be observed that, in general, Henry's Law shows higher residuals than the other models, which indicates a worse performance, especially in $H_2S$. In the same way, adjusting the value to Henry's constant does not improve the simulation performance. At the same time, the best results are obtained with the two-parameter model and extended UNIQUAC, depending on the case. On the other hand, it is confirmed that there is no substantive difference (*p*-values) between the four models at higher concentrations, even though the two-parameter model behaves better under these conditions.

From the results, it is possible to observe that the use of Henry's law for pure water resulted in important deviations because the culture medium has a significant amount of dissolved compounds which affect many variables in the liquid phase, such as the ionic strength. Using more accurate models, such as the extended UNIQUAC, improves the simulation results; however, its high complexity limits its use. The results of the semi-empirical models have the advantage of giving more adjusted results that can be obtained through experimental data from simple laboratory-scale experiments, making its use more convenient.

## 5. Conclusions

It has been established that using a two-parameter model provides good fits compared to Henry's law and extended UNIQUAC at lower concentrations, presenting significant deviation levels from experimental data. Considering the differences shown by the determinations, it is shown that the use of complex models with many parameters does not necessarily mean an improvement in the simulation results. On the other hand, in complex media, the oversimplification of constituents' effects could bring deviations that could be important depending on the accuracy required in such simulations. Therefore, using semi-empirical models can considerably improve the level of accuracy in simulations where there are complex interactions between the phases, using simple experimental systems to determine the necessary parameters

**Author Contributions:** Conceptualization, J.S.; methodology, G.A.; software, J.S.; validation, J.S., M.M.; formal analysis, J.S.; investigation, M.M.; resources, G.A.; data curation, J.S.; writing—original draft preparation, M.L.; writing—review and editing, R.O.-S.; visualization, R.O.-S.; supervision, J.S.; project administration, G.A.; funding acquisition, G.A. All authors have read and agreed to the published version of the manuscript.

**Funding:** National Agency of Research and Development (ANID) Ministry of Science Knowledge and Innovation of Chile, Project FONDECYT 1211569.

**Data Availability Statement:** Not applicable.

**Conflicts of Interest:** The authors declare no conflict of interest. The funders had no role in the design of the study, in the collection, analyses, or interpretation of data, in the writing of the manuscript, or in the decision to publish the results.

## Appendix A

Parameter description.

| Nomenclature | Description | Units |
|---|---|---|
| $a_s$ | Specific surface per reactor unit volume | $m^{-1}$ |
| A | Adjustable Debye–Hückel parameter | Nondimensional |
| B | Adjustable Debye–Hückel parameter | $kmol^{0.5}$ |
| $c_i$ | Molar concentration | M |
| $c_b{}^{in}$ | Biofilm concentration | ppm |
| $c_F$ | Inlet gas concentration | ppm |
| $c_g$ | Gas-phase concentration | ppm |
| $c_L$ | Liquid phase concentration | ppm |
| $C_b$ | Dimensionless concentration in the biofilm | Nondimensional |
| $C_g$ | Dimensionless concentration in the gas phase | Nondimensional |
| d | Liquid phase density | $g\,L^{-1}$ |
| EE | Expected value | Nondimensional |
| H | Biofilter height | m |
| $H_i$ | Henry's constant for species i | $M\,atm^{-1}$ |
| I | Ionic force | M |
| $K_s$ | Monod's constant | ppm |
| LSD | Least significant difference | Nondimensional |
| m | Partition coefficient | Nondimensional |
| MSE | Mean square error | Nondimensional |
| P | Pressure | atm |
| Pe | Péclet's mass transfer number | Nondimensional |
| PM | Molecular weight | $g\,mol^{-1}$ |
| $q_i$ | UNIQUAC's relative surface area | Nondimensional |
| $r_i$ | UNIQUAC's relative molecular volume | Nondimensional |
| R | Ideal gas constant | $atm\,L\,mol^{-1}\,K^{-1}$ |
| r | The spatial ordinate in the biofilm | m |
| RMSE | The root of mean square error | Nondimensional |
| T | Temperature | K |
| t | t-student statistic | Nondimensional |
| Ti | Residence and diffusion times ratio | Nondimensional |
| Th | Thiele modulus | Nondimensional |
| $u_{ij}$ | UNIQUAC's adjustable interaction parameter | Nondimensional |
| $v_b$ | Biofilm specific volume | Nondimensional |
| $V_g$ | Gas phase volume | L |
| $V_L$ | Liquid phase volume | L |
| $V_z$ | Gas velocity | $m\,s^{-1}$ |
| $X_b$ | Biomass concentration in biofilm | $g\,L^{-1}$ |
| Y | Input vector | Nondimensional |
| $Y_{X/S}$ | Biomass yield coefficient over TRS | $g\,g^{-1}$ |
| W | Dispersion coefficient | $m^2\,s^{-1}$ |
| z | Axial coordinate | m |
| Z | Element charge number | Nondimensional |
| **Superscript** | | |
| U | UNIQUAC (Universal Quasichemical) Model | - |
| DH | Debye–Hückel Model | - |
| $\alpha$ | The two-parameter model adjustment parameter | Nondimensional |
| $\alpha_i$ | Fraction of support surface covered by biofilm | Nondimensional |
| $\beta$ | The two-parameter model adjustment parameter | Nondimensional |

| Nomenclature | Description | Units |
|---|---|---|
| $\delta$ | Biofilm thickness | m |
| $\zeta$ | Dimensionless axial coordinate | Nondimensional |
| $\gamma$ | Activity coefficient | Nondimensional |
| $\kappa_i$ | Debye length of species i | $\text{Å}^{-1} \cdot \text{I}^{1/2} \cdot \text{m}^{-1}$ |
| $\kappa$ | Nondimensional specific degradation velocity | Nondimensional |
| $\mu_i$ | The root of the mean square error of model i | Nondimensional |
| $\mu_{max}$ | Specific growth velocity | $\text{s}^{-1}$ |
| $\phi_b$ | Nondimensional biomass concentration | Nondimensional |
| $\psi$ | Nondimensional special coordinate in the biofilm | Nondimensional |
| $\tau_R$ | Gas residence time | s |
| $\tau_R$ | Characteristic diffusion time in the biofilm | s |

**Appendix B**

Reactions considered in the extended UNIQUAC model.

$$H_2S_{(g)} \leftrightarrow H_2S_{(l)}$$

$$DMS_{(g)} \leftrightarrow DMS_{(l)}$$

$$MM_{(g)} \leftrightarrow MM_{(l)}$$

$$DMDS_{(g)} \leftrightarrow DMDS_{(l)}$$

$$CO_{2(g)} \leftrightarrow CO_{2(l)}$$

$$CO_{2(l)} + H_2O_{(l)} \leftrightarrow H_2CO_{3(l)}$$

$$H_2CO_{3(l)} \leftrightarrow HCO_3^{-}{}_{(l)} + H_{(l)}^{+}$$

$$HCO_3^{-}{}_{(l)} \leftrightarrow CO_3^{-2}{}_{(l)} + H_{(l)}^{+}$$

$$H_2O_{(l)} \leftrightarrow OH_{(l)}^{-} + H_{(l)}^{+}$$

$$Na_2HPO_{4(l)} \leftrightarrow 2Na_{(l)}^{+} + HPO_{4(l)}^{-2}$$

$$KH_2PO_{4(l)} \leftrightarrow K_{(l)}^{+} + H_{(l)}^{+} + HPO_{4(l)}^{-2}$$

$$MgCl_{2(l)} \leftrightarrow Mg_{(l)}^{+2} + 2Cl_{(l)}^{-}$$

$$(NH_4)_2SO_{4(l)} \leftrightarrow 2NH_{4(l)}^{+} + SO_{4(l)}^{-2}$$

$$MnCl_{2(l)} \leftrightarrow Mn_{(l)}^{+2} + 2Cl_{(l)}^{-}$$

$$FeCl_{3(l)} \leftrightarrow Fe_{(l)}^{+3} + 3Cl_{(l)}^{-}$$

$$Na_2CO_{3(l)} \leftrightarrow 2Na_{(l)}^{+} + CO_{3(l)}^{-2}$$

$$Na_2S_2O_{3(l)} \leftrightarrow 2Na_{(l)}^{+} + S_2O_{3(l)}^{-2}$$

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
