# Peer review of "Simulation of the Biofiltration of Sulfur Compounds: Effect of the Partition Coefficients"

_processes, doi:10.3390/pr10071325_

Round 1

Reviewer 1 Report

The study is relevant to the field of air purification biotechnology. I have some comments for the authors:

1. The annotation could include numerical values of the main results obtained during the modeling;

2. Write the diameter of the polypropylene ring in the methodology;

3. Not all elements (pump, valves, tanks ....) are numbered in Figure 1. It is not clear how the temperature is maintained;

4. To provide information on the basis of which the input concentrations (4, 10 and 20 ppm) were chosen. Why so low?

5. References to formulas are not everywhere in the text (Eq. 2, 3, 4, ...);

6. The simulation results could be compared with the results of foreign authors.

Reviewer 2 Report

“Total reduce sulfurs”. Not clear. Do you mean total reduced sulfur? Could you please check English language?

Abstract: Please provide the full definitions of H2S, DMS, MM, and DMDS compounds in the abstract (e.g., hydrogen sulfide (H2S), …).

There is no background information and concluding remark in the beginning and in the end of abstract, respectively.

Introduction:

Line 29: It is not clear what particular systems the authors are referring

In the introduction part, more information should be provided about different semi-empirical models and their applications in biofiltration studies.

The authors did not emphasize enough the scientific novelty of the study.

Materials and methods

More information is needed about operation conditions of biotrickling filter. For example, there is no information about inlet gas concentrations, gas temperature, air flow rate, empty bed residence time, etc.

How long the microorganisms were grown before the start of the experiment?

What were detection limits of studied contaminants in gas chromatograph?  

I suggest to describe Figure 1 in more detail.

Authors performed ANOVA test, however there is no section provided about statistical analysis in the Materials and methods.

Results

Figures 2, 3, 4: It is not appropriate to explain different lines (HA _.._), (TM __ __), etc. in Figure captions. Please add legends to all figures including explanation of black circles

Please describe the results shown in Figures 2-4

The results of Table 4 were also described only little. More details should be added.

The discussion made in the manuscript is very short. The authors also needs to add more discussion to the work.  Please, include implications of the study and future prospects

Reviewer 3 Report

This article aimed to simulate the gas-liquid partition coefficient of a bio-filtration system using different models and compare the results to experimental data, and hence, investigate the accuracy of the system simulation. When it comes to novelty, nevertheless, I do not see anything new or interesting in this paper. Additionally, there is a lack of discussions in this paper despite having a separate section, all we see here are some results without providing strong evidence and a relationship between them. I believe these features are essential for papers to be published in the Processes journal. 

Some specific comments: 

1.     the article was not prepared well and inaccuracies are obvious in the content and sentences, making the readers confused.

2.     The model studied at high concentrations seems to have very little accuracy, what is the reason for this error, and what can be done about it?

3.     Most of the references they used in their paper are old-fashioned! They should have done a decent literature review before starting this study.

4.     The introduction section should be reconsidered and improved to better reflect the aim of the study.

5.     The conclusion section should be reconsidered as it does not reflect all the stories in this paper.

Based on the above-mentioned comments the paper is not appropriate to be published in the Processes journal.

Round 2

Reviewer 2 Report

The authors have improved the manuscript and it can be accepted in the present form